# Are Oropharyngeal Dysphagia Screening Tests Effective in Preventing Pneumonia?

**DOI:** 10.3390/jcm11020370

**Published:** 2022-01-13

**Authors:** Ikuko Okuni, Satoru Ebihara

**Affiliations:** Department of Rehabilitation Medicine, Toho University Graduate School of Medicine, Tokyo 143-8541, Japan; pon1990@med.toho-u.ac.jp

**Keywords:** dysphagia, aspiration, screening, water swallowing test, Mann Assessment of Swallowing Ability, Gugging Swallowing Screen

## Abstract

Oropharyngeal dysphagia, a clinical condition that indicates difficulty in moving food and liquid from the oral cavity to the esophagus, has a markedly high prevalence in the elderly. The number of elderly people with oropharyngeal dysphagia is expected to increase due to the aging of the world’s population. Understanding the current situation of dysphagia screening is crucial when considering future countermeasures. We report findings from a literature review including citations on current objective dysphagia screening tests: the Water Swallowing Test, Mann Assessment of Swallowing Ability, and the Gugging Swallowing Screen. Pneumonia can be predicted using the results of the screening tests discussed in this review, and the response after the screening tests is important for prevention. In addition, although interdisciplinary team approaches prevent and reduce aspiration, optimal treatment is a challenging. Intervention studies with multiple factors focusing on the elderly are needed.

## 1. Introduction

Oropharyngeal dysphagia is a clinical condition characterized by difficulties in moving food and liquid from the oral cavity to the esophagus. The prevalence of oropharyngeal dysphagia is markedly high; 16% of elderly individuals aged 70–79 years who live independently suffer from oropharyngeal dysphagia, increasing to 33% in those aged 80 years or older. Moreover, 51% of the elderly aged 65 years or older living in institutions suffer from dysphagia [1]. Thus, oropharyngeal dysphagia has a markedly high prevalence. Furthermore, the prevalence of many diseases that can cause oropharyngeal dysphagia increases with age. Thus, it has been suggested that age-related changes contribute to oropharyngeal dysphagia. The international trend of aging shows that the proportion of people aged 65 years and older in the total population (population aging rate) has risen from 5.1% in 1950 to 8.3% in 2015, and it is expected to rise to 17.8% by 2060. Population aging will likely progress rapidly in the next half of this century [2]. As such, the number of elderly people with oropharyngeal dysphagia is expected to increase worldwide.

Oropharyngeal dysphagia can cause loss of appetite, malnutrition, and poor physical function; it can ultimately lead to life-threatening situations such as aspiration pneumonia and asphyxiation accidents. Therefore, understanding the current oropharyngeal dysphagia situation is crucial, especially when considering countermeasures for the future. In recent years, the recommendation to restart oral intake early to maintain swallowing function and shorten the treatment period, if possible, even in patients with acute aspiration pneumonia, has been advocated [3]. Swallowing function can be evaluated using a variety of methods, including interviews, medical examinations, screenings, and evaluative tests using specialized equipment. Through conversations with patients and their families, interviews systematically collect the information necessary for diagnosis, match this information with medical knowledge, and draw inferences to formulate and direct diagnosis; thus, interviews are an important first step in the care process. Specifically, interviews are essential for obtaining comprehensive medical histories and identifying symptoms suspected to cause oropharyngeal dysphagia, as well as systemic symptoms, such as nutritional and respiratory status. Symptoms associated with oropharyngeal dysphagia can be systematically identified using designated questionnaires. The accuracy of some of these questionnaires, including the Seirei Dysphagia Screening Questionnaire and the Eating Assessment Tool-10 (EAT-10), has been validated for dysphagia screening. Subjective screening questionnaires in the elderly may reduce the prevalence of oropharyngeal dysphagia, due to factors such as unawareness of swallowing problems, swallowing difficulties perceived as natural consequences of aging, and the presence of silent aspiration. Therefore, the use of subjective dysphagia screening in combination with objective dysphagia screening increases the prevalence of oropharyngeal dysphagia [4]. Ideally, such screening should yield a positive result for all individuals with oropharyngeal dysphagia (sensitivity) and a negative result for all individuals without it (specificity), which can be carried out without the use of special equipment. An ideal screening instrument is highly needed in medical institutions, long-term care facilities, nursing facilities, and visiting care settings where equipment is not available. Moreover, with an ideal screening test, unnecessary referrals and tests may be reduced. Furthermore, dysphagia screening should be able to prevent aspiration pneumonia. Here, we conducted a literature review on non-instrumental, objective dysphagia screening tests and outlined their effectiveness in preventing pneumonia.

## 2. Literature Search on Dysphagia Screening

In order to understand the current status of non-instrumental objective screening tests in recent years, we conducted a literature search to identify screening tests that have been suggested to be effective and have been used in multiple institutions (Figure 1).

In the first step, we defined the four elements of the clinical question as follows: population: elderly people with suspected dysphagia; intervention: non-instrumental screening tests; comparison: instrumental tests (fiberoptic endoscopic swallow study, videofluoroscopic swallow study); and outcome: effectiveness of the diagnosis of aspiration. This was then followed by a search for articles using a structural formula that combines the words “dysphagia”, “aspiration”, and “screening” screening in PubMed-yielded 2932 articles. We then narrowed down the article search to meta-analyses, randomized controlled trials, and systematic reviews published within the last 10 years (from January 2011 to July 2021), which yielded 102 articles. We tried to limit the number of papers to those that studied the elderly, but there were too few, so we instead excluded papers on single diseases only, excluding stroke. Furthermore, we excluded studies that used equipment such as fiberoptic endoscopic swallow studies and videofluoroscopic swallow studies, and those related to oral care, treatment, and training. Subsequently, 19 suitable articles were identified. A similar search and exclusion process using the Web of Science database resulted in 20 papers. After reading the articles, we excluded duplicates (*n* = 14), an article examining children (*n* = 1), those not related to the accuracy of screening tests (*n* = 8), and systematic reviews that failed to indicate the effectiveness of screening tests (*n* = 7). We additionally excluded a randomized controlled trial using the DREP screening (DREPs) [5], which is a protocol used in Brazil, and a randomized controlled trial using an original semi-solid swallowing test [6], as these are conducted in few facilities. We found three systematic reviews related to the water swallowing test [7,8,9], indicating that it is the most common screening test. Systematic reviews of effective swallowing screening for acute stroke [10] and swallowing screening in nursing homes [11] showed that the Gugging Swallowing Screen is a highly reliable tool with high sensitivity. In addition, a 2020 systematic review of the Gugging Swallowing Screen showed similar results [12], indicating that the Gugging Swallowing Screen is an effective screening test that can be used in different facilities. Perren et al. conducted a systematic review to evaluate post-extubation dysphagia in critically ill patients [13]. Despite the lack of available data on dysphagia screening in intensive care settings, they stated that the Mann Assessment of Swallowing Ability might serve as a reliable validated tool for diagnosing dysphagia in stroke patients. Based on these reports, we examined three dysphagia screening tests: the Water Swallowing Test, Mann Assessment of Swallowing Ability, and Gugging Swallowing Screen. The original articles discussed in the adopted reviews, including citations, are presented in Table 1. However, the articles were limited to those with a sample size of at least 40, and we divided the protocols for the Water Swallowing Test into three types (single sips, consecutive sips, and progressive amounts). Each of the three articles was excerpted in chronological order from the most recent to the oldest.

We reviewed the methodological quality of each included study, using criteria from the Quality Assessment of Studies of Diagnostic Accuracy (QUADAS-2) tool, as recommended by Cochrane (www.quadas.org; accessed on 12 December 2021). Results of the methodological quality assessment for each of the 15 included studies are shown in Table 2. We considered three studies to be at low risk across all four risk-of-bias domains: patient selection, index test, reference standard, and flow and timing (McCullough et al. [16], Trapl et al. [26], and Warnecke et al. [27]). Three studies were at low risk of bias for three domains (Hey et al. [20], Mann et al. [23], and Antonios et al. [25]). A major concern for risk of bias across the other included studies was that they had not enrolled consecutive patients or were unknown. In addition, some studies adopted a convenience sampling method. Finally, there were a number of high-risk studies due to the fact that they were carried out by people who knew the respective results of the index test and the reference standard. It was also unclear in eight studies whether there was an appropriate time interval between the index test and the reference standard. There were no applicability concerns for 15 studies across the three applicability domains: patient selection, index test, and reference standard.

## 3. Water Swallowing Test

The amount of water used in the Water Swallowing Test (WST) varies, with 3–20 mL for single swallowing, 90–100 mL for continuous swallowing, and 2–50 mL for a titration method, which increases the amount of water in stages, each exhibiting varying levels of diagnostic accuracy (Table 1). Among these stages, the test has the most suitable characteristics for excluding aspiration when a certain amount of water (e.g., 90 mL), is continuously ingested. In addition, the WST, with a single ingestion of a small volume, was superior for correctly classifying aspirating patients. Therefore, it may be possible to increase the sensitivity and specificity within a screening session of the same patient by performing both continuous drinking of a certain amount and single drinking of a small amount in stages [7]. However, there are currently no articles on the WST that support a high sensitivity and high specificity, and continuous swallowing of 90 mL of water is not recommended for patients who have begun to show signs of dysphagia or require a tracheostomy tube to secure the airway [20,29].

A recent prospective study of 102 patients aged 75 years or older (mean age 84.5 years) who were admitted to a geriatric ward reported a sensitivity of 76.6% and specificity of 65% (examiner: physician) for continuous swallowing of 90 mL of water [30]. In a cross-sectional study of 94 community-dwelling older people (aged 65 years and over) living independently, there was no significant difference in maximum tongue pressure between the sarcopenic (47.0 kPa) and non-sarcopenic (48.6 kPa) groups, but the time taken to drink 100 mL of water was significantly longer in the sarcopenic group (12.43 s) than in the non-sarcopenic group (5.66 s). This finding suggests that sarcopenic patients have a reduced ability to swallow [31]. The cut-off value for tongue pressure in the diagnostic criteria for sarcopenic dysphagia is less than 20 kPa [32]. This delay may be an early predictor of dysphagia before clinical problems become apparent.

### 3.1. Research Results on WST and Pneumonia

Most studies evaluating the effectiveness of the WST and pneumonia prevention have been conducted in patients with a single disease. Miki et al. conducted a symptom questionnaire, the Repetitive Saliva Swallowing Test (RSST), and the WST for dysphagia screening using a single 30 mL intake protocol in 85 postoperative patients with stomach cancer [33]. The results showed that postoperative pneumonia was not observed in patients who tested positive in the screening tests. Furthermore, the authors reported that intervention by the rehabilitation department to which the positive patients were referred was important in reducing the incidence of pneumonia. Three patients who developed postoperative pneumonia tested negative in the screening tests, and the cause of pneumonia was thought to be an aspiration in all cases. These results suggest that the screening tests used in this study did not sufficiently identify patients at high risk for aspiration pneumonia.

Ebersole et al. examined the incidence of hospital-acquired aspiration pneumonia (HAAP) in 12,392 hospitalized cancer patients who did or did not participate in nursing-initiated dysphagia screenings [34]. The incidence of HAAP per 1000 discharged patients who underwent dysphagia screening by the WST (continuous swallowing of 90 mL of water) was 8.78, and that per 1000 discharged patients who did not undergo dysphagia screening was 7.36. The study reported that dysphagia screening had no apparent effect on the incidence of HAAP. However, patients at high risk of oropharyngeal dysphagia, such as those with a history of head and neck cancer, were excluded from the screening process in this study; instead, they underwent more comprehensive instrumental swallowing evaluations. In addition, 30% of HAAP patients associated with difficulty in swallowing were fasting prior to aspiration, highlighting the difficulty of preventing HAAP in this population. Discontinuing an oral diet is not equivalent to eliminating the risk of HAAP. The authors stated that aspiration of secretions, microaspiration of oropharyngeal bacteria, and reflux associated with tube feeding were the causes of HAAP risk, regardless of dietary status.

Oguchi et al. conducted a retrospective study of 97 post-extubation patients undergoing cardiovascular surgery [35]. The endpoints were consciousness level (Glasgow Coma Scale), RSST, WST (3 mL in single intake), speech intelligibility score, and risk of dysphagia in the cardiac surgery score (RODICS) [36]. They reported that WST was the strongest predictive factor of postoperative pneumonia compared to other evaluations and that the incidence of pneumonia increased approximately three-fold when aspiration was suspected by the WST. However, 57.14% of patients who did not start oral intake, according to the results of the WST, were subsequently diagnosed with pneumonia. The authors stated that it is important to analyze the causal mechanism and consider measures to prevent pneumonia, such as improvement in wakefulness, swallowing function training, assistance with phlegm expulsion, airway suction, postural drainage, and oral care.

### 3.2. Combining the WST with Other Screening Tests

Surprisingly, there are few articles on the WST and pneumonia prevention. As the WST evaluates airway response and voice changes, it may overlook silent aspiration, which is caused by the absence of a cough reflex or throat ringing even when a substance is absorbed in the subglottis. To compensate for this oversight, pulse oximetry, cervical auscultation, and cough tests were used in a combined evaluation. Pulse oximetry, which measures oxyhemoglobin saturation in peripheral capillaries, is used to detect a decrease in saturation that suggests aspiration during swallowing. However, the diagnostic accuracy of pulse oximetry in predicting aspiration is controversial, and current evidence does not support its use [37].

Cervical auscultation determines dysphagia, mainly in the pharyngeal phase, by listening to swallowing and breathing sounds using a stethoscope placed on the neck. Although it has long been used as a non-invasive and common screening method, it lacks sufficient objectivity and reliability among available evaluators because of the limitations of the human auditory system and the fact that the stethoscope is designed and tuned for a specific purpose, such as observing heart or lung sounds [38]. However, a device called high-resolution cervical auscultation (HRCA) has been developed, which is expected to be applied clinically as a non-invasive screening method and as a biofeedback method during treatment [39].

We did not find any studies evaluating the effectiveness of the WST in combination with pulse oximetry or cervical auscultation for preventing pneumonia, but one study investigated the association between the combined WST and cough test, and the onset of pneumonia, which is shown below. Nakamori et al. investigated the association between the RSST, the WST (3 mL in single intake), and a cough test with the onset of pneumonia in acute stroke patients [40]. Each test was performed on 226 patients upon admission; the patients were then monitored for 30 days. Of these, 17 patients developed pneumonia during the observation period, and the sensitivity and specificity of the WST were 29.4% and 95.2%, respectively. Other screening tests alone did not adequately predict the risk of aspiration pneumonia. However, combining these three tests increased the sensitivity and specificity to 88.2% and 83.7%, respectively, demonstrating their usefulness for predicting aspiration pneumonia. Moreover, the authors stated that the risk of silent aspiration was thought to be high when an abnormality was found in the cough test, and strategies to prevent aspiration pneumonia, such as pulse oximetry, were crucial. Perry et al. conducted a randomized controlled trial of cough test for dysphagia in 311 acute stroke patients [41] and developed the Dysphagia in Stroke Protocol (DiSP), a standardized management protocol based on a previous study showing that pneumonia was not reduced in patients who underwent a cough test [42]. They then investigated the changes in clinical outcomes after using DiSP in patients with acute stroke (*n* = 432). DiSP is a protocol in which patients who have passed the cough test proceed to the evaluation of oral intake, and patients who have failed the cough test immediately undergo a videofluoroscopic swallow study (VFSS) without oral intake. The study results of the study showed that the incidence of aspiration pneumonia after DiSP was 10%, regardless of the cough test, which was significantly lower than the 28% observed prior to using DiSP. Nevertheless, the mortality rate of patients who developed pneumonia was only slightly reduced. Thus, to reduce pneumonia-related mortality, the authors concluded that proper management of patients with silent and dominant aspiration is more important than simply identifying patients with potential aspiration.

In summary, the WST achieves various levels of sensitivity and specificity as described in the literature, but it is well known that the WST may overlook silent aspiration as it evaluates by airway response and voice changes. Therefore, further research is needed to establish the most effective combination of screening tests to detect silent aspiration. In addition, management to avoid the onset of pneumonia, such as oral care, is important for preventing pneumonia.

## 4. Mann Assessment of Swallowing Ability

The Mann Assessment of Swallowing Ability (MASA) was developed by the American speech therapist Mann and colleagues to evaluate swallowing dysfunction in acute stroke patients [43]. As a clinical assessment tool rather than a screening test for dysphagia, the MASA can quantify the risk of aspiration in a bedside setting using the following 24 endpoints: general patient examination (alertness, cooperation, auditory comprehension, aphasia, apraxia, and dysarthria); the oral preparation phase (saliva, lip seal, tongue movement, tongue strength, tongue coordination, oral preparation, respiration, and respiratory rate for swallowing); the oral phase (gag reflex, palatal movement, bolus clearance, and oral transit time); and the pharyngeal phase (cough reflex, voluntary cough, voice, tracheostomy, pharyngeal phase, and pharyngeal response). Each endpoint of the MASA is evaluated on a scale of 5 or 10 points, with a total score of 200 points. A lower score for each endpoint indicates a higher severity of dysphagia, and the suspicion of dysphagia or aspiration can be determined from the total score of each endpoint. In acute stroke patients, dysphagia is suspected with a total MASA score of 177 points or lower, and aspiration is suspected with a total MASA score of 169 points or lower.

The sensitivity and specificity of the MASA for predicting of dysphagia in stroke patients were reported to be 73% and 89%, respectively, in comparison with VFSS. In addition, the sensitivity and specificity of the MASA for the prediction of aspiration were reported to be 93% and 63%, respectively [23]. The MASA is currently used for various diseases, and its sensitivity and specificity for evaluation in a mixed disease population were 39.6% and 59%, respectively, when VFSS was used as the gold standard [24]. Antonios et al. statistically reviewed the original MASA data and identified items important for developing a clinical assessment tool that could be used more rapidly and accurately [25]. As a result, they devised a modified MASA (mMASA), a simplified version of the MASA, to utilize highly distinguishable items. The mMASA showed a sensitivity of 87–92% and a specificity of 84.2–86.3% in predicting dysphagia with respect to the original MASA (Table 1).

As mentioned above, the MASA was developed to evaluate the eating and swallowing dysfunction in acute stroke patients, and the cut-off value for aspiration was set to 170 points. In addition, of the 24 items, 12 items of “alertness, cooperation, auditory comprehension, respiration, dysphasia, dysarthria, saliva, tongue movement, tongue strength, gag, voluntary cough, and palate” are specified in the mMASA. Therefore, a study examined the calculation of the cut-off value of the MASA suitable for the elderly requiring nursing care for various diseases and the usefulness of the endpoints of the MASA [44]. In this study, based on the total score of the MASA and the results of the fiberoptic endoscopic swallow study (FEES), a cut-off value for aspiration was 122 points with a sensitivity and specificity of 75% and 90%, respectively. In addition, the sensitivity and specificity were 90% and 33%, respectively, when the cut-off value of 170 points for acute stroke patients was used. This indicates that the number of false-negative diagnoses increased when the original cut-off value was used for the elderly requiring nursing care. In addition, of the 24 items evaluated in the MASA, the following 8 items were shown to be potentially useful for evaluating the eating and swallowing functions of the elderly requiring nursing care: 6 items of “cooperation, oral preparation, oral transit time, cough reflex, pharyngeal phase, and pharyngeal response” that do not require the execution of instructed movements, as well as 2 items of “tongue strength and tongue coordination” which have been associated with sarcopenic dysphagia [31,45].

### Research on MASA and Pneumonia

The MASA has been used to predict pneumonia in hospitalized patients. Mitani et al. conducted a retrospective observational study to determine whether the onset of pneumonia could be predicted in 393 hospitalized patients (average age 79.2 ± 11.4 years). The etiologies of the participants were as follows: Parkinson’s disease (*n*  =  111), multiple cerebral infarctions (*n*  =  73), cerebral infarction (*n*  =  57), orthopedic diseases (*n*  =  23), disuse (*n*  =  10), cerebral hemorrhage (*n*  =  9), subarachnoid hemorrhage (*n*  =  6), chronic obstructive lung disease (*n*  =  3), neuromuscular disease (excluding Parkinson’s disease; *n*  =  66) and others (*n*  =  35)) [46]. The follow-up period was 365 days, and the items of the MASA, Functional Independence Measure (FIM), and Controlling Nutrition Status (CONUT) were investigated. The results showed that 102 patients developed pneumonia and that FIM and MASA scores were significantly lower in the group with pneumonia than in the group without pneumonia, while the average age and the CONUT scores were higher in the group with pneumonia than in the group without pneumonia. The cut-off MASA score was 170.5 points, with a sensitivity and specificity of 70% and 83%, respectively, and the authors stated that the MASA might be a useful tool for predicting the onset of pneumonia. Chojin et al. conducted a prospective cohort study of 153 elderly hospitalized patients with pneumonia who were evaluated by a speech-language pathologist using the MASA (average age of 85.4 ± 9.9 years) [47]. In this study, a multivariate analysis showed that a MASA score of 169 points or lower was an independent risk factor for recurrence of pneumonia within 30 days and mortality after 6 months. Therefore, the authors stated that, for patients with a low MASA score, it is important to start oral care and appropriately evaluate and support various aspects of life, such as diet, posture during meals, mealtime, and degree of care.

The MASA can be performed with minimal items, such as a penlight, a tongue depressor, and test foods that require mastication. With its low invasiveness, the MASA can be performed by speech therapists and nurses, in addition to doctors, and is considered useful for temporal evaluations, such as regular evaluation to predict the onset of pneumonia and evaluation of training effects. In patients with a low MASA score, it is important to prevent pneumonia by intervening early in dysphagia. However, the MASA, which focuses on indirect endpoints of swallowing, requires additional tests to propose a food style suitable for the patient.

## 5. Gugging Swallowing Screen

One of the screening tests that can recommend a food style is the Gugging Swallowing Screen (GUSS), which was developed at the Landesklinikum Donauregion Gugging in collaboration with Danube University Krems in Austria to evaluate the severity of dysphagia and the risk of aspiration in acute stroke patients. The GUSS is currently used to evaluate dysphagia in various diseases, and it has been translated into multiple languages and is widely used internationally [48]. The GUSS is divided into an indirect swallowing test in Part 1 and a direct swallowing test in Part 2, consisting of three subtests, all of which must be performed in succession. The direct swallowing test starts with semisolid foods, which are considered less challenging for acute stroke patients, and gradually step up to more challenging test items, such as liquid and solid test foods. In addition, the liquid swallowing subtest of the GUSS uses a titration method in which the amount of water is gradually increased in steps of 3, 5, 10, 20, and 50 mL. The subtests are evaluated based on points, with higher points indicating a better grade. A maximum of five points is given in each subtest, with a total of 20 points over four subtests, and a patient must achieve the maximum score of 5 points to advance to the next subtest. Based on the score, the following diet is recommended: a regular diet for the maximum score of 20 points, a swallowing-adjusted diet and small amounts of liquid for 15–19 points, a baby food-like swallowing-adjusted diet in combination with an alternative nutrition method for 10–14 points, and no oral intake for 9 points or lower. In addition, consultation with a swallowing specialist and further evaluation by FEES and VFSS is recommended for patients with 19 points or lower.

Studies on the effectiveness of the GUSS have been carried out on stroke patients. In addition, the cut-off value of the GUSS was set to 14 points, and it was examined using FEES. Trapl et al. prospectively evaluated 50 acute stroke patients in an upright position of at least 60 degrees in bed and capable of recognizing the examiner’s face, spoon, and texture in front of him/her at the start of GUSS [26]. They found that the GUSS had a sensitivity of 100%, a specificity of 50–69%, and a negative predictive value of 100%. Warnecke et al. examined 100 acute stroke patients using a prospective, double-blind method and reported that the GUSS screened for risk of aspiration with a high sensitivity of 96.5% and a specificity of 55.8% [27]. They also stated that low specificity was associated with the high rate of failure to complete the initial part of the GUSS in severe cases with a National Institute of Health Stroke Scale score of 15 points or higher. The effectiveness of the GUSS may vary depending on the severity of the stroke. Said Bassiouny et al. prospectively evaluated 40 acute stroke patients [28]. High sensitivity (93.7%) and high specificity (92.5%) were observed in patients who were clearly conscious and able to follow instructions (Table 1).

### Research on GUSS and Pneumonia

Studies relating to GUSS and pneumonia have evaluated measures for predicting stroke-associated pneumonia (SAP) in stroke patients and their role in reducing the incidence of SAP. Quyet et al. prospectively surveyed 508 patients hospitalized within 5 days of stroke onset, which showed an incidence of SAP of 13.4% [49]. Logistic regression analysis showed that a GUSS score of 15 points or lower was associated with SAP (odds ratio 11.7, 95% confidence interval of 6.6–20.8, *p*-value < 0.01). The authors concluded that dysphagia was an independent risk factor for pneumonia. Dang et al. conducted a cohort study enrolling 892 acute stroke patients, which showed an incidence of SAP of 13.8% [50]. With a sensitivity and specificity of 80.5% and 80.1%, respectively, they stated that the GUSS was superior in predicting SAP. In addition, logistic regression analysis showed an odds ratio (OR) of 11.4, a 95% confidence interval of 7.4–17.5, and a *p*-value of < 0.01 (solid food is bread).

Regarding the decrease in the incidence of SAP, Teuschl et al. compared patients evaluated by the GUSS with those who were not evaluated, using a database containing 1394 patients hospitalized for acute stroke [51]. A total of 993 patients (71.2%) was screened by GUSS, of whom 50 (5.0%) developed SAP. The incidence of SAP in these patients was 22 (5.5%), which was comparable to that in 401 patients who were not screened. The incidence of SAP was low compared to the overall incidence in recent meta-analyses, suggesting that the GUSS was effective in preventing SAP. However, no difference was observed between the two groups, which may be because cases of extremely severe stroke, for which early testing was not possible, and very mild cases, for which pneumonia was not expected, were likely not screened, masking the positive effects of the intervention. Furthermore, they concluded that identifying patients at risk for SAP using the GUSS, including those with very mild stroke, and other management factors, such as the timing of nasogastric tube insertion, oral hygiene, and administration of antibiotics, in addition to diet therapy, were also helpful in further reducing the incidence of SAP. Sørensen et al. conducted a controlled trial in acute stroke patients, which consisted of an intervention group that received oral hygiene based on GUSS by a speech and language pathologist, a standardized care plan immediately after admission (*n* = 58), and a control group that received arbitrary clinical swallowing screening and oral hygiene (*n* = 88) [52]. The results showed that the incidence of SAP was 7% in the intervention group and 27% in the control group, and they reported that the incidence of SAP was significantly reduced by early systematic screening using the GUSS and enhanced oral hygiene (solid food is dry bread). Similarly, another study compared the incidence of SAP between the intervention group (*n* = 186), in which nurses performed GUSS all year round, and the control group (*n* = 198), in which only speech therapists performed GUSS during working hours [53]. They reported that the time from admission to GUSS was shorter in the intervention group and that the incidence of SAP was significantly lower in the intervention group (3.8%) than in the control group (11.6%). Furthermore, the intervention group had shorter hospital stays and lower short-term in-hospital mortality rates. However, another study compared the duration of systematic 10-mL WST (*n* = 204) with the duration of systematic GUSS administration by trained nurses (*n* = 140), reporting that they showed no difference of SAP or mortality [54].

Based on these results, SAP can be predicted and prevented by systematically implementing the GUSS at an early stage and examining the food style. This allows for early evaluation of the swallowing function after hospitalization, and prompt consultation with a swallowing specialist or performance with FEES and VFSS. Upon inspection of dysphagia or aspiration, the incidence of SAP is thought to be reduced. Further studies, including those on patients other than those with acute stroke, are expected to validate the effectiveness of the GUSS. A 2012 systematic review by Wilkinson et al. of bedside diagnostic tests for aspiration and predictors of pneumonia in older patients without stroke stated that the existing evidence is insufficient to support the use of bedside tests in the general elderly population A systematic review of predictors of inflammatory bowel disease stated that existing evidence was insufficient to support the use of bedside testing in the general population [55]. Currently, a scoping review protocol on the psychometric properties of tools for initial screening of oropharyngeal dysphagia in older people is underway [56].

## 6. Conclusions

We believe that pneumonia can be predicted by screening tests, such as the WST, MASA, and GUSS, as discussed in this review. Studies that reported the prevention of pneumonia used these screening tests in combination with DiSP, early systematic GUSS administration, or enhanced oral hygiene. Therefore, the response after predicting pneumonia by screening tests at an early stage is thought to be important for preventing pneumonia. Many studies have stated that an interdisciplinary team approach improves the efficiency and quality of treatment to prevent and reduce aspiration. However, it is difficult to identify which team approach is most effective and which type of treatment combination is the most optimal. Future intervention studies that investigate multiple factors and focus on the elderly are needed.

## Figures and Tables

**Figure 1 jcm-11-00370-f001:**
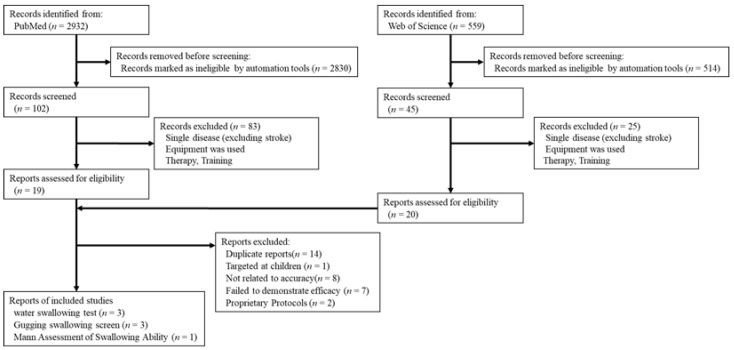
Study selection flowchart.

**Table 1 jcm-11-00370-t001:** Dysphagia screening methods (sensitivity and specificity).

Study (Year)	Patients Included (Etiology)	Swallowing Assessment Method	Inspector	Reference Test	Outcome	Sensitivity	Specificity
		Water Swallowing Test					
Wakasugi et al. [14]. (2008)	107 (MD)	Single sips 3 mL, Total amount 3 mL	NR	VFSS, FEES	Aspiration	69	97
Momosaki et al. [15]. (2013)	110 (CVA)	Single sips 4 mL, Total amount ≤12	SLP	FEES	Aspiration	93	79
McCullough et al. [16]. (2005)	165 (CVA)	Single sips 5 mL, Total amount ≤10	Doctor	VFSS	Aspiration	44	94
		Single sips 10 mL, Total amount ≤20	Doctor	VFSS	Aspiration	37	96
Suiter et al. [17]. (2008)	3000 (MD)	Consecutive sips 90 mL	Doctor	FEES	Aspiration	97	49
Zhou et al. [18]. (2011)	107 (CVA)	Consecutive sips 90 mL	Doctor	VFSS	Aspiration	87	42
Patterson et al. [19]. (2011)	126 (HNC)	Consecutive sips 100 mL	NR	FEES	Aspiration	80	77
Hey et al. [20]. (2013)	80 (HNC)	Progressive amounts 2, 5, 10, 20 mL Total amount ≤51 mL	SLP	FEES	Aspiration	100	61
Somasundaram et al. [21]. (2014)	67 (CVA)	Progressive amounts 5, 10, 20 mL Total amount ≤50 mL	SLP	VFSS	Aspiration	70	81
Hassan et al. [22]. (2014)	74 (MD)	Progressive amounts 5, 20, 50 mL Total amount ≤75 mL	NR	FEES	Aspiration	74	70
Mann et al. [23]. (2000)	128 (CVA)	MASA	SLP	VFSS	Swallowing disorder	73	89
González-Fernández et al. [24]. (2011)	133 (MD)	MASA	SLP	VFSS	Aspiration	39.6	59
Antonios et al. [25]. (2010)	150 (CVA)	Modified MASA	Doctor	MASA	Swallowing disorder	87~92	84.2~86.3
Trapl et al. [26]. (2007)	50 (CVA)	GUSS	SLP Nurse	FEES	Aspiration	100	50~69
Warnecke et al. [27]. (2017)	100 (CVA)	GUSS	SLP	FEES	Aspiration	96.5	55.8
Said Bassiouny et al. [28]. (2017)	40 (CVA)	GUSS	SLP	FEES	Aspiration	93.8	96.1

CVA: cerebrovascular accident, HNC: head and neck cancer, MD: mixed-disease. SLP: speech-language pathologist, NR: not reported. VFSS: videofluoroscopic swallow study, FEES: fiberoptic endoscopic swallow study; MASA: Mann Assessment of Swallowing Ability, GUSS: Gugging Swallowing Screen. All solids in the GUSS were dry bread.

**Table 2 jcm-11-00370-t002:** Assessment of the quality of the included studies.

Study (Year)	Risk of Bias				Applicability Concerns		
	Patient Selection	Index Test	Reference Standard	Flow and Timing	Patient Selection	Index Test	Reference Standard
Wakasugi et al. [14]. (2008)	-	-	?	?	+	+	+
Momosaki et al. [15]. (2013)	-	+	?	+	+	+	+
McCullough et al. [16]. (2005)	+	+	+	+	+	+	+
Suiter et al. [17]. (2008)	?	-	+	-	+	+	+
Zhou et al. [18]. (2011)	?	-	-	?	+	+	+
Patterson et al. [19]. (2011)	?	+	-	?	+	+	+
Hey et al. [20]. (2013)	?	+	+	+	+	+	+
Somasundaram et al. [21]. (2014)	+	+	?	?	+	+	+
Hassan et al. [22]. (2014)	?	+	-	?	+	+	+
Mann et al. [23]. (2000)	+	+	+	?	+	+	+
Gonzá-lez-Fernández et al. [24]. (2011	+	-	+	?	+	+	+
Antonios et al. [25]. (2010)	+	-	+	+	+	+	+
Trapl et al. [26]. (2007)	+	+	+	+	+	+	+
Warnecke et al. [27]. (2017)	+	+	+	+	+	+	+
Said Bas-siouny et al. [28]. (2017)	?	+	+	?	+	+	+

(+): Low Risk, (-): High Risk, (?): Unclear Risk. The risk of bias and applicability were assessed according to QUADAS-2 (www.quadas.org, accessed on 12 December 2021).

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
