# Peer review of "Are Oropharyngeal Dysphagia Screening Tests Effective in Preventing Pneumonia?"

_jcm, 2022, doi:10.3390/jcm11020370_

Round 1

Reviewer 1 Report

I thank you for your editing, I do have some few suggestions

1) Line 81 , you should add the inclusion criteria base on PICO (Population: age, diagnosis...Intervention: presence of dysphagia; Comparison: type of article, for example case series, case-control, rct...

2) Masa is not a screening but a clinical assessment tool, which is complete different assessment. I would suggest to clarify this in the manuscript. 

Author Response

Thank you for your comments. We have answered each of your points below. For manuscript changes, we used Microsoft Word's "Track Changes" feature. Please note the number of lines is the number of lines when all the changes/comments in the Track changes are displayed.

Comments and Suggestions for Authors

I thank you for your editing, I do have some few suggestions

 1) Line 81 , you should add the inclusion criteria base on PICO (Population: age, diagnosis...Intervention: presence of dysphagia; Comparison: type of article, for example case series, case-control, rct...

The purpose of our literature search was to identify screening tests that have been suggested to be effective and are being used at multiple sites, and to learn about the recent implementation of non-instrumental, objective screening tests. We therefore defined the four elements of the clinical question as follows: Population: elderly people with suspected dysphagia; Intervention: non-instrumental screening tests; Comparison: instrumental tests (fiberoptic endoscopic swallow study, videofluoroscopic swallow study); Outcome: effectiveness of the diagnosis of aspiration, then We conducted a literature search with meta-analyses, randomized controlled trials, and systematic reviews over the past 10 years. We also tried to limit the number of papers to those that studied the elderly, but there were too few, so we instead excluded papers on single diseases, excluding stroke, and those on children. Furthermore, we excluded studies that used equipment such as fiberoptic endoscopic swallow studies and videofluoroscopic swallow studies, and those related to oral care, treatment, and training. After reading the articles, we made reference to tests that were non-surgical, objective, demonstrably valid (predictive of aspiration), and used in a multi-center setting. We then investigated whether these screening tests could prevent pneumonia.

Accordingly, we have added the following to lines 74-77 in the revised manuscript.

The first step, we defined the four elements of the clinical question as follows: Population: elderly people with suspected dysphagia; Intervention: non-instrumental screening tests; Comparison: instrumental tests (fiberoptic endoscopic swallow study, videofluoroscopic swallow study); Outcome: effectiveness of the diagnosis of aspiration.

"Types of articles: meta-analyses, randomized controlled trials, and systematic reviews" is as in the previous version of the manuscript. (Lines 78 to 79 in the revised manuscript)

2) Masa is not a screening but a clinical assessment tool, which is complete different assessment. I would suggest to clarify this in the manuscript.

Thank you for pointing this out. We have made the following corrections (Line 258 in the revised manuscript).

Correction from “As a screening test for dysphagia” to “As a clinical assessment tool rather than a screening test for dysphagia” (Line 278 in the revised manuscript).

Correction from” a screening tool” to “a clinical assessment tool”

We hope the revised version is now suitable for publication and look forward to hearing from you in due course.

Reviewer 2 Report

In this manuscript (jcm-1550386), the authors revised the first version.
These revisions may be accessible to many readers. I judge that this MS
generally reaches the publication standard, but it is necessary to provide
a supplementary explanation for the findings of Table 2 which the authors
added.

Author Response

Thank you for your comments. We have answered each of your points below. For manuscript changes, we used Microsoft Word's "Track Changes" feature. Please note the number of lines is the number of lines when all the changes/comments in the Track changes are displayed.

Comments and Suggestions for Authors

In this manuscript (jcm-1550386), the authors revised the first version.

These revisions may be accessible to many readers. I judge that this MS generally reaches the publication standard, but it is necessary to provide a supplementary explanation for the findings of Table 2 which the authors added.

Thank you for pointing this out. We have added the following text to lines 111 to 125 in the revised manuscript.

"We reviewed the methodological quality of each included study, using criteria from the Quality Assessment of Studies of Diagnostic Accuracy (QUADAS‐2) tool, as recommended by Cochrane (www.quadas.org). Results of the methodological quality assessment for each of the 15 included studies are shown in Table 2. We considered three studies to be at low risk across all four risk of bias domains: patient selection, index test, reference standard, and flow and timing (McCullough et al [16], Trapl et al [26], Warnecke et al [27]). Three studies were at low risk of bias for three domains (Hey et al [20], Mann et al [23], Antonios et al [25]). A major concern for risk of bias across the other included studies was that they had not enrolled consecutive patients or were unknown. In addition, some studies adopted a convenience sampling method. Finally, there were a number of high-risk studies due to the fact that they were carried out by people who knew the respective results of the index test and the reference standard. It was also unclear in eight studies whether there was an appropriate time interval between the index test and the reference standard. There were no applicability concerns for 15 studies across the three applicability domains: patient selection, index test, and reference standard."

We hope the revised version is now suitable for publication and look forward to hearing from you in due course.

This manuscript is a resubmission of an earlier submission. The following is a list of the peer review reports and author responses from that submission.

Round 1

Reviewer 1 Report

Screening and assessment tools in dysphagia have been done since  1980s and sometime wrongly performed for the diagnosis of dysphagia. Recently the real differences between this methods have been considered correct among researchers. The study presented has conceptual flaws, methodological weaknesses and, disregards that the professional practice in dysphagia differs internationally as explained below:

1-About the arguments to build a strong research -  Operational Definition must be completely correct: considering that the authors constructed a research question based on three operational definitionss (elderly people, screening and evaluation, and dysphagia), it is essential that their definitions about these be clear and robust. When there is a generalization of the studied population, the authors do not struggle in their bibliographical search for healthy elderly people (members of normal aging) of elderly people with neurological and other diseases, they increase the risk of bias in the study, generating inappropriate conclusions. Furthermore, the concept of dysphagia mentioned by the authors and described in the first line of the introduction should be associated with the description of the dysphagic picture present also at the beginning of the second paragraph. What is the definition of the diagnosis of dysphagia for the authors in the context of the study?. In addition, and more serious in the view of this reviewer, it deals with the misapplication of the concept of screening and assessment in dysphagia used by the authors and still very connected to the controversial understanding, and in great current debate, of these methods built in the 80's. Screening is just a method can any health professional can apply to identify risk of dysphagia, however assessment is a specialized method to evaluated all aspects of swallowing and should be appply by an specialist in swallowing who are differet in distints countries. Besides this, I strongly do not recommend that the authors whrite that swallowing assessment or even an interview is solely a medical procedure as the authors wrote. In many countries the swallowing assessment is a procedure done  by speech language pathologyst, nurses or dentistry and not just by medical doctors. Specialized practice in dysphagia depends on the origins of training in the area of each country and must be respected.

  2-Study design:  Unfortunately, the method chosen for this simple narrative review is not robust enough to answer the research question in aim. Scoping reviews require would be the appropriate study design with other method if the authors' intended conclusion can be made. Furthermore, I emphasize that the Water Swallowing Test and GUSS are only screenings, while the MASA test is currently one of the assessment tool with best psychometric validity.

Reviewer 2 Report

Comments to the Authors
In this manuscript (jcm-1474434), the authors conducted a literature review on dysphagia screening tests, WST, MASA, and GUSS. They concluded that pneumonia can be predicted and prevented by these tests and interdisciplinary team approaches prevent and reduce aspiration. This MS shows comprehensive results, but I have some concerns about this MS:

Major comments

1. Study selection

I understood the search method of the six literature articles based on Figure 1, but it is unclear how the authors selected the 15 articles in Table 1. Are the 15 articles in Table 1 the cited literature of the six review articles? If it is so, the authors should expressly state it not only in “Literature Search on Dysphagia Screening” but also in the abstract.

2. Food style

In Table 1, the amount of water used in WST was shown, but the types of food used in MASA and GUSS were not described. Information about the solid test food especially used in GUSS is helpful for readers. Although the solid food likely varies between countries, please add the information in Table 1.

Reviewer 3 Report

I would like to thank for the opportunity to revise this interest and important manuscript.

Nevertheless, there are some revisions which should be edited before publication:

1) I would suggest the author to follow the PRISMA guideline check-list ( https://www.equator-network.org/reporting-guidelines/prisma/) as the methods for systematically review the literature are different.

2) The manuscript lacks to explicit mention the inclusion/exclusion criteria

3) The authors should analyse the quality of the study, for example using  QUADAS-2

4) The authors use only one database (pubmed) but a systematic review must include other database such as web of science.....

I suggest to make these revisions.

Author Response

Please see he attachment.
